# Exploring Obscurin and SPEG Kinase Biology

**DOI:** 10.3390/jcm10050984

**Published:** 2021-03-02

**Authors:** Jennifer R. Fleming, Alankrita Rani, Jamie Kraft, Sanja Zenker, Emma Börgeson, Stephan Lange

**Affiliations:** 1Department of Biology, University of Konstanz, 78457 Konstanz, Germany; 2Centre for Molecular and Translational Medicine, The Wallenberg Laboratory and Wallenberg, Department of Molecular and Clinical Medicine, University of Gothenburg, 41345 Gothenburg, Sweden; alankrita.rani@wlab.gu.se (A.R.); jamie.kraft@wlab.gu.se (J.K.); 3Department of Medicine, University of California, San Diego, CA 92093, USA; sanja.zenker@uni-bielefeld.de; 4Department of Clinical Physiology, Sahlgrenska University Hospital, 41345 Gothenburg, Sweden

**Keywords:** obscurin, striated muscle enriched protein kinase, SPEG, kinase

## Abstract

Three members of the obscurin protein family that contain tandem kinase domains with important signaling functions for cardiac and striated muscles are the giant protein obscurin, its obscurin-associated kinase splice isoform, and the striated muscle enriched protein kinase (SPEG). While there is increasing evidence for the specific roles that each individual kinase domain plays in cross-striated muscles, their biology and regulation remains enigmatic. Our present study focuses on kinase domain 1 and the adjacent low sequence complexity inter-kinase domain linker in obscurin and SPEG. Using Phos-tag gels, we show that the linker in obscurin contains several phosphorylation sites, while the same region in SPEG remained unphosphorylated. Our homology modeling, mutational analysis and molecular docking demonstrate that kinase 1 in obscurin harbors all key amino acids important for its catalytic function and that actions of this domain result in autophosphorylation of the protein. Our bioinformatics analyses also assign a list of putative substrates for kinase domain 1 in obscurin and SPEG, based on the known and our newly proposed phosphorylation sites in muscle proteins, including obscurin itself.

## 1. Introduction

The obscurin protein family consists of three members: the giant protein obscurin with its three main splice isoforms (obscurin-A, obscurin-B and obscurin associated kinase KIAA1639), striated muscle enriched protein kinase (SPEG) with its two major muscle splice isoforms (SPEGα and SPEGβ), and obscurin-like 1 (Obsl1) [1,2,3,4,5,6,7,8,9]. Two of the family members—obscurin and SPEG—possess domains that link the proteins to cellular signaling pathways. The giant obscurin isoforms contain a calcium/calmodulin-binding IQ-motif and RhoGEF domain triplet (consisting of serially linked SH3-DH-PH domains). Obscurin-B has an additional two C-terminally located kinase domains (OK1 and OK2), which are also present in the obscurin-associated kinase isoform [2,6,10]. This tandem kinase domain architecture is also found in SPEG—containing the kinase domains SK1 and SK2, which are homologous to kinase domains found in obscurin [7].

Obscurin and SPEG kinases have been classified to belong to the myosin light-chain kinase (MLCK) family, while also showing considerable similarity to death-associated protein kinases (DAPKs; 58% similarity), titin kinase (54% similarity) as well as invertebrate twitchin (54% similarity) [7]. However, the activity of these diverse groups of kinases is differently regulated. DAPK and MLCK are regulated by calcium/calmodulin-binding [11] and other peptide motifs, such as the PEF/Y-motif that is located within the catalytic domain of DAPK [12], while twitchin and titin contain a pseudosubstrate located C-terminally to the catalytic domain [13,14,15]. Sequence identity with known active kinases also indicates that all kinase domains within obscurin and SPEG have well-conserved residues important for coordination of ATP and enzymatic function [5,10,15,16]. This seems to be in opposition to the invertebrate obscurin and SPEG orthologue unc-89, where only kinase domain 2 may be capable of acting as a true protein kinase, while kinase domain 1 is thought to be a pseudokinase [16,17,18].

Several reports suggest and verify that the mammalian obscurin and SPEG kinases are catalytically active [5,10,15,16,19]. Identified substrates and binding partners for SPEG include the ryanodine receptor (RyR2), junctophilin-2 (Jph2), the sarcoplasmic/endoplasmic reticulum calcium ATPase2 (Serca2), or the phosphatase myotubularin (Mtm1) [20,21,22,23]. Mammalian obscurin kinases OK1 and OK2 were shown to interact and potentially phosphorylate N-cadherin, Fhl2/Dral or titin [24], while the kinase domains in the invertebrate unc-89 isoforms bind to a complex containing Lim-9/Fhl and the protein phosphatase Scpl-1 (small CTD phosphatase-like-1; Scp) [17,18,25]. Further analysis of this interaction identified two binding sites for the Lim-9/Scpl-1 complex: a region surrounding OK1, and another portion of the protein surrounding OK2, which also contains part of the low sequence complexity inter-kinase region [18]. Intriguingly, the minimal binding sites for the invertebrate phosphatase Scpl-1 and the mammalian phosphatase Mtm1 have considerable overlap, perhaps suggestive of functional conservation throughout evolution.

The substantial number of interaction partners that are involved in calcium cycling and homeostasis, as well as sarcoplasmic reticulum architecture and function, indicates crucial functions for SPEG and obscurin in cardiac and skeletal muscles (reviewed in [9,26,27,28,29]). Indeed, loss of function models for SPEG and identified human mutations have been linked to a number of cardiomyopathies. SPEG knockout mice develop severe cardiomyopathy and heart failure [20,30]. Patients with pathological gene-variants of SPEG show a range of symptoms depending on the location of the mutation (reviewed in [31]). For example, several individuals suffering from centronuclear myopathy that also exhibited dilated cardiomyopathy or hypertension carried either homozygous or compound heterozygous mutations in SPEG, including at least two patients with mutations in SK2 [22,32]. In addition, insufficient levels of SPEG protein in patients suffering from cardiac arrhythmias may contribute to pathologically altered diastolic calcium handling and the development of paroxysmal atrial fibrillations [23].

Comparatively, little is known about the role that obscurin plays in hearts. Cardiac stress, such as aortic constriction, was identified as a potent regulator of the *obscurin* gene expression, linking increased obscurin levels to hypertrophic growth and adaptive remodeling of the heart [33]. In addition, several human *obscurin* gene variants have been linked to the development of hypertrophic cardiomyopathy, dilated cardiomyopathy, left ventricular noncompaction or arrhythmogenic right ventricular cardiomyopathy [28,34], all of which underscore the importance of obscurin for proper cardiac function. Surprisingly though, unlike SPEG knockouts, loss of obscurin in mice is tolerable [35], suggesting that loss of function can be compensated during cardiac development. Part of the explanation may be due to overlapping functions between obscurin and other members of this protein family [36]. More recently, a new mouse model was established that sheds further light on the biological functions of obscurin: mice that delete Ig-domains 58/59 of the protein display age-dependent pathological remodeling of the heart and arrhythmias [37].

Few data are available that investigate the kinase domains in obscurin and SPEG, their biology, regulation and substrate specificity. Data from in vitro assays revealed that SK1 is capable of autophosphorylation as well as substrate phosphorylation without the aid of calcium/calmodulin (CaM) [5,20]. However, analysis of the peptide sequence following SK1 nonetheless suggests the presence of a calcium/CaM-binding motif [15]. Besides the autoregulatory and putatively calcium/CaM-dependent mechanism, SPEG is also controlled by other cellular kinases. Two kinases known to phosphorylate SPEG are calmodulin-dependent protein kinase II (CaMKII) [38] and protein kinase B (PKB), which was identified as a master regulator for SPEG kinase activity [39]. PKB phosphorylates SPEG in response to insulin at three distinct sites (Ser2461, Ser2462 and Thr2463) within the inter-kinase region of the protein. Phosphorylation of SPEG at these sites activates SK2, which in turn modulates Serca2 phosphorylation and activity [39]. Intriguingly, SK2 does not seem to contain any easily discernible autoregulatory elements [15].

Similar to SK1, the low sequence complexity inter-kinase region C-terminal to OK1 contains several predicted regulatory elements, including a CaM-binding motif (aa 6739-6752) that is embedded in the predicted autoinhibitory domain (aa 6724-6786) [10,15]. Furthermore, compared to the equivalent kinase domain in SPEG, OK2 is not predicted to contain similar regulatory motifs [10], although it was shown to undergo autophosphorylation when expressed and purified from insect cells [24]. These differences suggest that the modulation of OK1 and OK2 kinase activity may differ considerably [10].

We studied obscurin and SPEG kinase biology, specifically focusing on kinase domain 1 of both proteins. We found that the inter-kinase region C-terminal to OK1 contains several phosphorylation sites, while the same region in SK1 remained unphosphorylated. Homology modeling substantiates that OK1 contains all key amino acids important for its catalytic activity in the right locations. Our mutational analysis and molecular docking studies reveal further that OK1 autophosphorylates its adjacent low sequence complexity inter-kinase region, perhaps representing another mechanism for its regulation. Using a bioinformatics approach, we also predict possible SPEG and obscurin kinase domain 1 substrate based on the known and newly proposed phosphorylation site.

## 2. Materials and Methods

### 2.1. Cloning and Generation of Constructs

For the generation of the mouse obscurin kinase (OK, NM_001171512) and striated muscle enriched protein kinase (SK, NM_007463) mammalian constructs, total murine cardiac mRNA was transcribed to cDNA using Superscript IV (Thermo Fisher Scientific, Waltham, MA, USA) using random hexamers according to the manufacturer’s instructions. The following oligonucleotides were used to generate amplicons that were subsequently cloned in-frame into the pH A-C1 [40] expression vector (Table 1).

Site-directed mutagenesis was done as previously described [40] using an oligonucleotide, as shown in Table 1.

Correct in-frame integration of amplicons into the vector backbone and mutagenesis was verified by sequencing.

### 2.2. Cell Culture

African green monkey Cos-1 kidney cells and the mouse myoblast cell line C2C12 were purchased from ATCC (Manassas, VA, USA) (catalogue numbers CRL-1650 and CRL-1772, respectively), and cultured as previously described [40,41,42,43]. For transfection of DNA into Cos-1 or C2C12 cells, a mixture of 1 µg DNA, 100 µL DMEM and 3 µL lipofectamine-2000 (Life Technologies, Carlsbad, CA, USA) was added to the cells growing in 35 mm dishes or 6-well plates at 60–80% confluency. For extraction of proteins, cells were grown for another 24 h before lysis. For immunofluorescence of C2C12, cells were switched to differentiation medium 24 h after transfection and grown for another 2 or 7 days, with daily medium changes.

### 2.3. Immunofluorescence and Microscopy

Transfected and differentiated C2C12 cells were washed with 1× phosphate-buffered saline (PBS; Thermo Fisher Scientific, Waltham, MA, USA), followed by incubation with 4% paraformaldehyde (Sigma-Aldrich, Hamburg, Germany) dissolved in 1× PBS for 10 min at room temperature. After fixation, cells were permeabilized with 0.2% Triton-X100 (Sigma-Aldrich, Hamburg, Germany) in 1× PBS for 5 min. Following permeabilization, cells were incubated with a mixture of primary antibodies diluted into gold buffer (GB; 20 mM Tris-HCl, pH 7.5, 155 mM NaCl, 2 mM ethylene glycol tetra-acetic acid, 2 mM MgCl_2_, 5% bovine serum albumin (BSA); all chemicals from Sigma-Aldrich, Hamburg, Germany) for 2 h, washed three times for 5 min in 1× PBS, incubated with secondary antibodies diluted in GB for 1 h and washed three times again for 5 min in 1× PBS. For visualisation of filamentous actin and cell nuclei, Alexa-647 phalloidin (Thermo Fisher Scientific, Waltham, MA, USA) and DAPI (Sigma-Aldrich, Hamburg, Germany) were mixed with the secondary antibodies. After staining, cells were mounted into a fluorescent mounting medium (ProLong Gold anti-fade reagent; Life Technologies, Carlsbad, CA, USA) and processed for imaging on a Leica SP5 confocal microscope in sequential scanning mode, using a 63× oil-immersion objective and a zoom rate of 2. Images were analyzed using NIH-ImageJ (version 1.48; National Institutes of Health, Bethesda, MD, USA) and the BioFormats Importer (version 5.0.7; University of Dundee & Open Microscopy Environment, Dundee, UK), as well as Photoshop (version CS5; Adobe, San Jose, CA, USA).

### 2.4. Antibodies

To detect the expression of HA fusion constructs, the following antibody was used: anti-HA (Roche, Basel, Switzerland; catalogue number 11867423001). The generation and specificity of the obscurin antibody have been described elsewhere [35,36]. Secondary antibodies were either from Jackson ImmunoResearch (West Grove, PA, USA) or Cell Signaling (Danvers, MA, USA).

### 2.5. Protein Analysis

For expression and phosphorylation analysis of constructs, cells were washed in 1× PBS and lysed into 1× sample buffer (10% glycerol, 60 mM Bis-Tris/HCl pH 6.8, 2% SDS, 0.0006% bromophenol blue; all chemicals from Sigma-Aldrich, Hamburg, Germany). Samples were run on SDS-PAGE gels or 10% acrylamide bis-tris-gels (Biorad, Philadelphia, PA, USA) containing 10 µM Phos-tag-acrylamide (Wako Chemicals, Richmond, VA, USA; Cat. No. AAL-107; [44]) supplemented with 20 µM ZnCl_2_ (Sigma-Aldrich, Hamburg, Germany). After separation of proteins, gels were transferred as previously described onto nitrocellulose membranes [45].

After transfer, membranes were stained with Ponceau solution (Sigma-Aldrich, Hamburg, Germany) and imaged after de-staining using a Bio-Rad GelDoc imager (Bio-Rad, Philadelphia, PA, USA). Membranes were then blocked in blocking solution (Tris-buffered Saline solution supplemented with 0.2% Tween-20 (TBST) containing 5% BSA and 5% non-fat skim milk; all chemicals from Sigma-Aldrich, Hamburg, Germany) for 2 h at room temperature. Following blocking, membranes were incubated with primary antibody dissolved into blocking solution and incubated over night at 4 °C on a shaking platform. After incubation, membranes were washed in TBST three times for 10 min each at room temperature, incubated in blocking solution supplemented with appropriate secondary antibodies and incubated at room temperature for 1 h. After washing six times for 10 min each, membranes were developed using SuperSignal West Pico (Thermo Fisher Scientific, Waltham, MA, USA) and imaged on a Bio-Rad GelDoc Imager (Bio-Rad, Philadelphia, PA, USA). Uncropped original immunoblot images are available in Appendix A.

### 2.6. Modeling and Bioinformatics Analyses

Comparative modeling was performed using RosettaCM (version 3.12, Rosetta Commons, Johns Hopkins University, Baltimore, MD, USA) through the Rosetta server [46]. Five models were produced, either representing the closed or open states of the kinase. The model with the highest degree of similarity to the closed state was selected for further analysis. To aid visual orientation, ATP (mol2 file downloaded from Zinc15 [47]) was modeled into the active site using AutoDock Vina (version 1.1.2, The Scripps Research Institute, La Jolla, CA, USA) [48]. The top-scoring model of 3 docking attempts was used. It was identical in all docking attempts. Figure generation and further analysis were performed using UCSF Chimera (version 1.15, UC San Francisco, San Francisco, CA, USA) [49]. Structural alignments were performed using the “MatchMaker” tool within Chimera and surface electrostatics were calculated using the “Coulombic Surface Coloring” tool, which calculates the surface charge of underlining residues using Coulomb’s law. The following settings were used: distance-dependent dielectric constant (ε) of 4.0; a solvent-accessible surface cut-off of 1.4 Å; histidine residues implicitly protonated, and default grid settings.

A 42 residue peptide (amino acids 6770–6811) encompassing the putative phosphorylation site Ser6789 within the human obscurin-B inter-kinase region was modelled ab initio using C-Quark (version 1.0, University of Michigan, Ann Arbor, MI, USA) [50,51]. Five models were generated, of which model 1 with the highest C-score of −1.13 was used for structural comparison.

Secondary structure prediction was done using Jpred (version 4, University of Dundee, Dundee, UK) [52]. The alignment of proteins was computed in Multalin (version 5.4.1, INRAE, Toulouse, France) [53] using the Blosum62-12-2 comparison table. Determination of domain borders and motifs was calculated using SMART (version 9.0, EMBL, Heidelberg, Germany) [54,55].

Molecular phylogenetic analysis was conducted using MEGA7 (version 7, Megasoftware, The Pennsylvania State University, State College, PA, USA) [56]. The evolutionary history was inferred by using the maximum likelihood method based on the JTT matrix-based model [57]. The tree with the highest log likelihood (−2283.15) is shown. Initial tree(s) for the heuristic search were obtained automatically by applying Neighbor-Join and BioNJ algorithms to a matrix of pairwise distances estimated using a JTT model, and then selecting the topology with a superior log likelihood value. The tree is drawn to scale, with branch lengths measured in the number of substitutions per site. The analysis involved 4 amino acid sequences. All positions containing gaps and missing data were eliminated. There were a total of 247 positions in the final dataset.

Search for putative SK1 and OK1 phosphorylation sites within the human proteome was done using the Motif Search tool on GenomeNet website of the Kyoto University Bioinformatics Center. 

## 3. Results

### 3.1. Obscurin Contains Phosphorylation Sites C-Terminal to Kinase Domain 1

Recently, a phosphorylation site within the ryanodine receptor (RyR2; Ser2368 in human RyR2, Ser2367 as well as Ser2368 in mouse RyR2 [58]) was identified that was unequivocally phosphorylated by SPEG [23]. However, it remains undetermined which of the two kinase domains within SPEG is responsible for this catalytic activity. We wondered if SPEG or the highly homologous obscurin contain putative phosphorylation sites that are similar to SPEGs phosphorylation site within RyR2. Alignment of the peptide sequence surrounding the phosphorylation sites in human and mouse RyR2 suggested the presence of putative phosphorylation sites in the low sequence complexity inter-kinase regions C-terminal of obscurin-B kinase 1 (OK1) and SPEG kinase 1 (SK1) (Figure 1a). Specifically, both inter-kinase regions within SPEG and obscurin have a set of serine residues that align in good agreement with the target serines in human and mouse RyR2 (red arrows in Figure 1a).

To investigate if SPEG and obscurin kinases contain phosphorylation sites, we cloned the N-terminal kinase domains of mouse SPEG or mouse obscurin-B without or with their adjacent C-terminal inter-kinase fragment (termed SK1 and OK1, or SK1-ps and OK1-ps, respectively; Figure 1b). We transfected the resulting HA-tagged constructs into Cos-1 cells and harvested protein lysates 24 h after transfection. Analysis of protein phosphorylation showed no band-shift of the minimal kinase fragment (SK1) or the fragment encompassing part of the adjacent inter-kinase region (SK1-ps) in immunoblots of Phos-tag gels when compared to standard SDS-PAGE (Figure 1c). However, the study of the homologous kinase fragments in obscurin-B identified several bands in OK1-ps (P1-P3) that were shifted compared to the unphosphorylated P0 state (Figure 1d), which only makes up ~43% of all expressed OK1-ps. Analysis of the band intensities also suggests a low abundance of phosphorylation states P1 and P3, while phosphorylation state P2 represents the majority of modified OK1-ps (~39%). Similar to SK1, the fragment encoding for OK1 alone showed no band-shift in Phostag gels, suggesting that all phosphorylation sites are within the inter-kinase region of obscurin. Further truncation of this part of obscurin-B that eliminates the putative phosphorylation site suggested by alignment with RyR2 reduced the number of phosphorylated bands to one for OK1-ΔS7 (P1; Figure 1e). In addition, the band intensity of the OK1-ΔS7 P1 band is weaker compared to the P2 band in OK1-ps, which represents the major phosphorylation state of the OK1-ps fragment. This result suggests that the stretch of seven serine residues present in the OK1-ps fragment (eight serine residues in human obscurin-B) may be responsible for the major phosphorylation (P2 state) seen in OK1-ps.

In summary, the inter-kinase region following kinase domain 1 in obscurin contains several phosphorylation sites. Our data also suggest that the catalytic domains themselves (OK1 and SK1) are void of phosphorylation sites in the tested conditions.

### 3.2. Presence of OK1 Phosphorylation Sites Affects the Localization of the Fragment in Differentiated Muscle Cells

What are the biological effects of the phosphorylation sites within the inter-kinase region of obscurin? We tested the localization of HA-tagged OK1 or OK1-ps in undifferentiated myoblasts or differentiated C2C12 myotubes (Figure 2). Both obscurin fragments localized diffusely in the cytoplasm of undifferentiated C2C12 cells, and were excluded from the nucleus (Figure 2a). While OK1-ps maintained this localization in differentiated myotubes, OK1 was now also found in myonuclei of myotubes (arrowheads in Figure 2b, left panel), perhaps reflective of its altered biological activity. Notably, however, is that both fragments did not associate with myofibers by themselves, but may require the remainder of the protein for their proper localization to sarcomeric structures [1,24].

### 3.3. Modelling Suggests That OK1 Is an Active Protein Kinase

Alignment of the OK1 sequence with sequences of kinases where structural information are available suggested a high degree of identity (>30%) to death-associated kinases (DAPK), twitchin kinase, myosin light-chain kinase (MLCK) and calcium/calmodulin-dependent kinase (CaMK) with E-values ranging between 4e-45 and 9e-41. The high similarity to several known kinase structures allowed for comparative modelling of OK1 using RosettaCM [46]. Five models were produced. Upon inspection, the differences between these models were due to the kinase being modeled in open or closed conformations. Of these models, the model most similar to a closed (active) state of DAPK1 bound to an ATP analog was selected for further analysis (Figure 3a). The RMSD between the selected homology model and an example closed kinase (PDB accession number: 1JKL; [62]) was 0.943 Å between 238 pruned atom pairs, and therefore this model is in good agreement with a closed state kinase. Our homology model suggests that OK1 is not a pseudokinase in line with previous observations [10,15,16,19,24]. Alignment with the DAPK1 sequence also indicates that amino acids shown to be important for coordination of ATP and Mg^2+^-ions [19] are largely conserved between DAPK1 and OK1 (Figure 3b). Independent secondary structure analysis using Jpred [52] suggests a close overlap between DAPK1 and OK1 and validated conservation of catalytically important amino acids. The only exception is Ile160 in DAPK1, which has been replaced in OK1, OK2 and SK1 with a cysteine. SK2 has several notable alterations from important amino acids conserved in DAPKs. Among them is a change of the “Ala-x-Lys” motif to “Val-x-Lys”, which is also observed in a subset of known active kinases [16], and a more conservative change of Ile160 to valine. However, all of these changes did not influence the complexion of ATP, suggesting that all kinase domains within obscurin and SPEG are predicted to be active kinases.

### 3.4. Actions of OK1 Auto-Phosphorylate the Obscurin Inter-Kinase Region

Many kinases contain regulatory elements, such as autophosphorylation and pseudosubstrate sites. We wondered whether some of the phosphorylation sites located within the obscurin inter-kinase region following kinase domain 1 may be phosphorylated by catalytic actions of the adjacent kinase domain or if the phosphorylation is mediated by other kinases present in the cellular context. To determine if OK1 is an active kinase, we mutated Lys6591 (Lys6497 in human obscurin-B) of the conserved “Ala-x-Lys” motif to alanine, effectively generating a kinase-dead mutant. Analysis of the phosphorylation pattern of the OK1-ps-KA mutant showed loss of the major modified bands in immunoblots of Phos-tag gels (Figure 3c). This suggests that OK1 is an active protein kinase, which in addition is responsible for the majority of phosphorylation of its adjacent inter-kinase region within obscurin. The remaining phosphorylation of OK1-ps-KA may originate from other kinases present in the cellular context.

Summarized, actions of OK1 phosphorylate the adjacent inter-kinase region in obscurin, indicating that obscurin may follow similar kinases that contain pseudosubstrates or autophosphorylation sites, which are important for the regulation of their activity.

### 3.5. Structural Analysis of the Putative Autophosphorylation Substrate C-Terminal of OK1

To gain further insight into the biology of the putative autophosphorylation site, we investigated the structure of a 42 amino acid peptide centered around the first available serine modified by OK1 actions. Ab initio modeling of the peptide using C-Quark generated five models, all of which are predicated on containing two helices connected by a chain of the eight serines, which is without distinct secondary structure (Figure 3d). We selected a model with the highest C-score for additional analysis. Further modeling and comparison with structures of known kinase substrates (shown in Figure 3e as occupied by the inhibitor tails of twitchin (orange) and CaMK (light-blue) and the inhibitory substrate of PKA1 (lime-green)) suggested that the N-terminal helical segment fits into the peptide-binding region. The helix could fit in this groove surface, allowing for Ser6789 to be placed towards the kinase active site. Intriguingly, the human and mouse sequences diverge considerably in this region (Figure 3d, lower panel). The equivalent amino acid to Ser6789 in mouse obscurin-B (Ser6883) is replaced by alanine. This change is similar to the change observed in the putative phosphorylation site within SK1, which showed a change to leucine (Leu1918) at this position (Figure 1a). This result suggests some flexibility when it comes to the choice of target serine (Ser6789 or Ser6790 in human obscurin-B) for phosphorylation by OK1, also supported by the finding that both, Ser2367 and 2368 are found to be phosphorylated in mouse RyR2 [58]. However, these findings also indicate that the difference in the SK1 autophosphorylation substrate region (Leu1918 in humans and Leu1923 in mouse SPEG instead of a serine residue) should be inconsequential and not responsible for the observed lack of phosphorylation seen in Figure 1c.

Should the peptide adopt a helical conformation N-terminal of the stretch of serines, as predicted by several C-Quark models, then the basic residues become oriented together on one side of the helix (blue surface, left panel Figure 3f), creating a complimentary basic surface to the acidic peptide-binding grove (Figure 3f, red surface, right panel). Thus, the predicted peptide conformation compliments the shape, length and electrostatic charge of the modeled obscurin peptide-binding groove.

## 4. Discussion

The premise of our experiments and modeling was to further extend the knowledge about SPEG and obscurin kinase biology. Our data also add evidence to the emerging roles that the inter-kinase regions in both proteins may play in the modulation of kinase activity.

Our experimental results indicate that there are no major phosphorylation sites within the catalytic domains of SK1 and OK1. However, available data in Phosphosite [58] suggest the presence of at least one phosphorylation site close to the catalytic domains of OK1 (Thr6560 in humans) and SK1 (Ser1598 in human SPEG, Ser1603 in mouse SPEG), all with unknown biological roles. The reason for the absence of any phosphorylation sites within the catalytic domain may be owed to the cell-model we used to express OK1 and SK1. Modification of the catalytic domains themselves is most likely due to other cellular kinases. However, protein levels of kinases and their associated regulatory proteins, as well as their catalytic activity, vary greatly depending on the tissue type and external factors (e.g., activity of cellular signaling pathways). Both obscurin and SPEG are enriched in muscle tissues and only marginally expressed or absent from most non-muscle tissues. Hence, analysis of the phosphorylation state of the catalytic domains in cardiomyocytes or differentiated skeletal muscle may give further insights if our initial result on the absence of OK1 and SK1 phosphorylation is owed to the Cos-1 cells used in our experiments. We also cannot exclude that our truncation mutants are not properly folded, which may account for the lack of autophosphorylation of the catalytic domains of SK1 or OK1.

The main finding of our study was that the inter-kinase region immediately C-terminal of OK1 harbors at least three uncharacterized phosphorylation sites. The majority of the sites are dependent on the catalytic activity of OK1, as mutational analysis using a kinase-dead mutant reduced the number of detectable phosphorylation sites to one. Although our data strongly suggest that OK1 is able to autophosphorylate obscurin in this region, we cannot completely rule out that some of the sites may be modified by other kinases present in the cellular context. In addition, the biological function and exact identity of these phosphorylation sites remain uncertain. While our data and experiments suggest that Ser6789 or Ser6790 in human obscurin-B may be the target for OK1, a stretch of additional serine residues immediately C-terminal to Ser6790 may be modified instead (or in addition to Ser6789/6790). Moreover, alignment with the mouse version of obscurin suggested that only the following Ser6790 is evolutionarily conserved (Figure 3d). It remains unclear why SK1 was unable to modify a homologous motif in the inter-kinase region within SPEG. Perhaps SPEG requires modification by other cellular kinases that are not present in Cos-1 cells before SK1 becomes catalytically active. The finding that our kinase-dead OK1-ps-KA mutant retained single phosphorylation may support this idea. It also remains to be determined if the autophosphorylation of obscurin at this motif acts as a pseudosubstrate that modifies the activity of OK1. Available data on the biological functions of other phosphorylation sites with the inter-kinase region of SPEG support this possibility.

Several lines of evidence suggest that the inter-kinase regions in obscurin and SPEG play crucial roles in regulating the kinase activities within both proteins. This region in both proteins harbors a number of known phosphorylation sites. The best-studied are three PKB phosphorylation sites (Ser2461, Ser2462 and Thr2463) within the C-terminal part of the SPEG inter-kinase region prior to the Ig-domain [39]. Another kinase that is known to modify SPEG in this region is CaMKII. Injection of isoproterenol in mice resulted in the changed cardiac phosphorylation status of SPEG at Ser2135 (Ser2130 in humans) [38]. Moreover, isoproterenol reduces SPEG mRNA levels in mouse hearts, possibly contributing to ensuing heart failure phenotype and pathologically altered calcium handling [21]. While the exact biological roles of modifying SPEG at this serine remain uncharacterized, phosphorylation of SPEG by PKB crucially regulated the Serca2 phosphorylation status by modulating SK2 activity [39], suggesting similar functions for other residues located within the inter-kinase region.

Obscurin remains less characterized than the comparatively better-studied SPEG protein. However, databases such as Phosphosite, suggest that obscurin is also phosphorylated within the low sequence complexity inter-kinase region. Perhaps the best-studied and most evolutionary conserved modification occurs at Ser6773 (Ser6867 in mice) just prior to the putative OK1 autophosphorylation site within obscurin (Figure 1a) [59,60]. This site was also found in a phosphoproteome study to identify proteins modified downstream of β1-adrenergic receptor signaling. Motif analysis suggests that Erk/Mapk kinases are responsible for its modification [61]. However, no biological function has been assigned to this phosphorylation.

The activity of these kinases may also be regulated by other structural elements and motifs, such as the putative autoinhibitory or CaM-binding regions [10,15], which also await further analysis. Indeed, one of the truncation constructs (OK1-ΔS7) used in our studies retains these elements. While this fragment loses the majority of observed phosphorylation sites, it retains a single strong P1 band in the Phos-tag gel (Figure 1e). It is an intriguing possibility that the remaining phosphorylation site should be located within the autoinhibitory region (spanning aa 6724–6786 in human obscurin-B, and also, including the CaM-binding site) as the catalytic domain of OK1 does not display discernible phosphorylation (Figure 1d).

Besides modulating kinase activity, data from a recent manuscript by Hiroshi Qadota, Guy Benian and their collaborators demonstrate that the low sequence complexity inter-kinase region in unc-89 is able to act as an entropic spring element. The mechanical properties together with the regulatory functions of the inter-kinase region open the intriguing possibility for a mechanosensory regulation of the kinases [67], e.g., exposure of inaccessible crucial regulatory phosphorylation sites by mechanical “stretching” of the protein may lead to activation of one or two kinases within unc-89. However, to date, this possibility is not supported by experimental evidence either in the nematode unc-89 or the mammalian SPEG and obscurin proteins. Nonetheless, this study also highlights the crucial importance of the inter-kinase region for muscle development and function, as animals with deletion mutants show severe defects in sarcomere organization, locomotion and force generation. Indeed, myopathies in patients with nonsense and frameshift mutations found in obscurin or SPEG [28] lend additional credence to this finding and suggest a high degree of functional evolutionary conservation.

It remains unclear if there is overlap in the protein substrates for the tandem kinase domains in SPEG and obscurin. However, looking at the phylogenetic tree of human kinases in both proteins suggests a closer homology of SK1 to OK1 and SK2 to OK2 than of the kinases within each protein (Figure 4a). These differences indicate further that there may be overlap in the choice of substrates for SK1 and OK1, as well as SK2 and OK2, respectively. Fortunately, phosphorylation sites in two proteins have been assigned to catalytic actions of SPEG kinases: Thr484 in Serca2 and Ser2368 in RyR2 [21,23]. While the identity of the kinase responsible for phosphorylating RyR2 remains unknown, Thr484 was shown to be modified by the catalytic actions of SK2. There is surprisingly very little similarity between the peptide sequences surrounding the modified serine and threonine residues in RyR2 and Serca2 (Figure 4b). As Serca2 is modified by SK2, one could reasonably speculate that RyR2 is modified by SK1. This hypothesis is strengthened by our studies, which indicate that a phosphorylation site in obscurin that contains similar elements to the motif found in RyR2 is modified by OK1. However, it remains to be experimentally verified if SK1 is indeed the kinase responsible for RyR2 phosphorylation.

Assigning a phosphorylation site unequivocally to a specific kinase opens up the possibility to search for similar phosphorylation motifs in other proteins. We wondered what other muscle proteins might contain phosphorylation sites, which have a peptide motif that is putatively recognizable by OK1 or SK1. While RyR2 modification by SK1 awaits experimental evidence, we used “Motif Search” and a degenerate peptide motif to search for human proteins containing matching phosphorylation sites (Figure 4b). Our bioinformatics analysis resulted in 923 possible kinase substrates (Appendix A). The list of proteins contains, besides the known SPEG substrates RyR2 and Jph2, also several known interaction partners of obscurin (e.g., Ank2 [68,69,70,71]), calcium channel subunits (e.g., Cacnb3, Cacna1c, Cacna1g, Cacna1h, Cacna1i), myosin heavychain isoforms (e.g., Myh1, Myh2, Myh8), the receptor tyrosine-protein kinase Erbb3, or Obsl1associated proteins (e.g., cullin-7, cullin-9 or Ahnak [72,73]). However, many of the identified sites show no or only poor evolutionary conservation. Further experimental evidence is needed to properly establish and/or refine the kinase substrate recognition motifs for obscurin and SPEG tandem kinases.

## Figures and Tables

**Figure 1 jcm-10-00984-f001:**
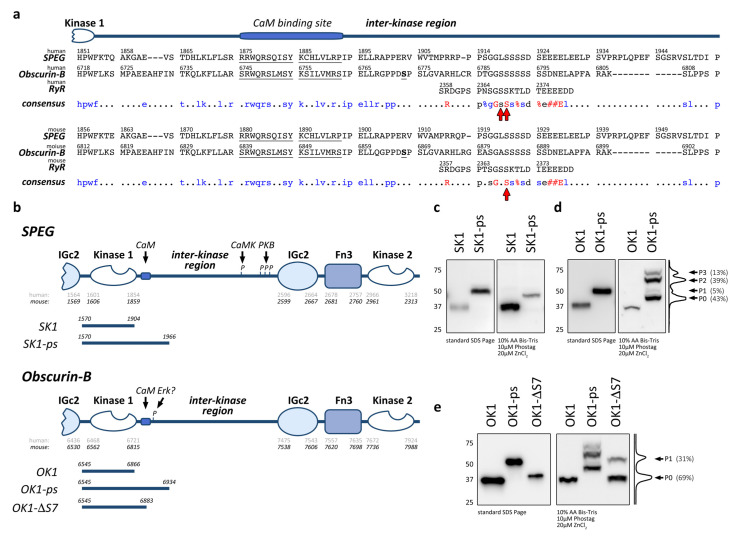
(**a**) Sequence alignment of striated muscle enriched protein kinase (SPEG) and obscurin inter-kinase regions following kinase domain 1, with the phosphorylation sites in human or mouse RyR2 highlighted by red arrows. The top panel shows the human sequences of SPEG (NM_005876.5), obscurin-B (NM_001098623.2) and RyR2 (NM_001035.3), while the bottom panel depicts the alignment of the respective mouse sequences (SPEG: NM_007463, obscurin-B: NM_001171512 and RyR2: NM_023868). The consensus sequences are shown with identical residues found in all three sequences highlighted in red, and sequences identical in SPEG and obscurin-B or the obscurin-associated kinase splice isoform highlighted in blue; # = D or E, % = S or T. The predicted calmodulin binding sites in human and mouse obscurin and SPEG [10,15] as well as the serine found to be putatively modified by Erk/Mapk in obscurin [59,60,61], are underlined. (**b**) Schematic overview of human and mouse SPEG and obscurin-B C-terminal domain organization. Residue numbers identify domain borders as determined by SMART. Location of the predicted calmodulin (CaM) binding sites [10,15] in SPEG and obscurin, mapped PKB and CaMK phosphorylation sites in SPEG [38,39], as well as the mapped putative Erk phosphorylation site in obscurin, are indicated [59,60,61]. The size and boundaries of mouse expression fragments used in this study are shown below the schematic domain organization. (**c**) Immunoblots of standard SDS-PAGE and 10 µM Phos-tag gels analyzing HA-tagged SK1 and SK1-ps expressed in Cos-1 cells. (**d**) Immunoblots of standard SDS-PAGE and 10 µM Pho-stag gels analyzing HA-tagged OK1 and OK1-ps expressed in Cos-1 cells. Histogram of the OK1-ps lane in the Phos-tag immunoblot and phosphorylation states (P0-P3) with band intensities (in%) are indicated. (**e**) Immunoblots of standard SDS-PAGE and 10 µM Phostag gels analyzing HA-tagged OK1, OK1-ps, and OK1-SΔ7 expressed in Cos-1 cells. Histogram of the OK1-ΔS7 lane in the Phos-tag immunoblot and phosphorylation states (P0-P1) with band intensities are indicated.

**Figure 2 jcm-10-00984-f002:**
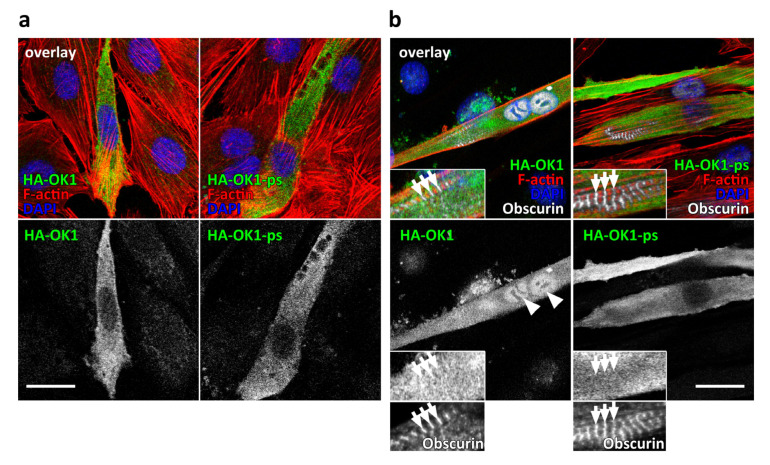
(**a**,**b**) Subcellular localization of HA-tagged OK1 and OK1-ps (green in the composite overlay and shown as a separate channel) in undifferentiated C2C12 myoblasts (**a**, three-channel composite overlay) or myotubes, differentiated for 7 days (**b**, four-channel composite overlay). Cells were counterstained with fluorescently linked phalloidin to visualize F-actin and DAPI (red and blue in the overlays, respectively). Proper differentiation of C2C12 was evaluated by sarcomeric localization of endogenous obscurin using an antibody directed against the IQ-64 region of the protein (arrows; white in the composite overlay of panel b, also shown as a separate channel in higher magnification below). The observed nuclear enrichment of OK1 in differentiated myotubes is highlighted by arrowheads (in b). Scale bars = 20 µm.

**Figure 3 jcm-10-00984-f003:**
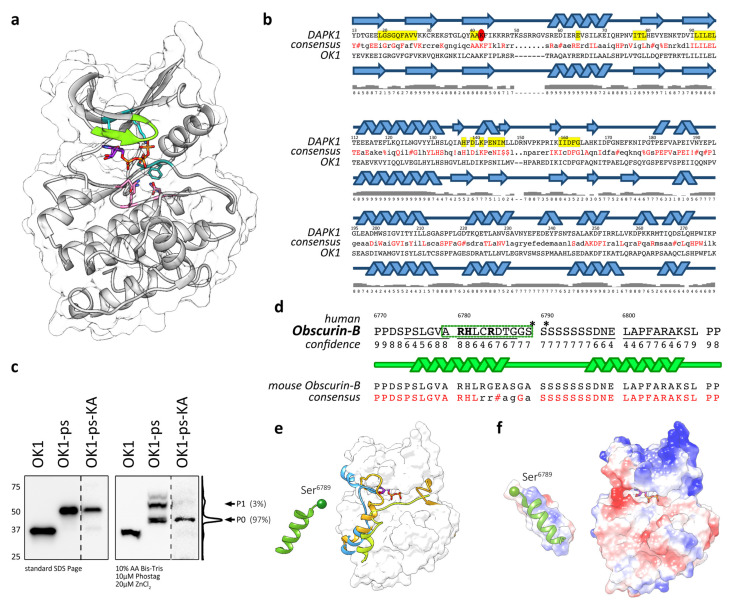
(**a**) Homology model of OK1 with residues required for catalysis. Conserved residues and motifs essential for kinase activity [19] are represented as sticks and carbons colored by motif as follows: G-rich loop (Gly-x-Gly-x-x-Gly) green; ATP binding motif (Ala-x-Lys) cyan; catalytic motif (His-x-Asp-x-Lys-x-x-Asn) pink; DFG motif (Asp-Phe-Gly) orange; ATP, purple. All other atoms are colored as follows: P, orange; O, red; N, blue. (**b**) Secondary structure comparison between DAPK1 and OK1 with amino acids known to be important for catalytic activity in DAPK1 highlighted in yellow [63]. Confidence of the predicted secondary structure is shown below the cartoon illustration. (**c**) Immunoblots of standard SDS-PAGE and 10µM Phos-tag gels analyzing the HA-tagged OK1, OK1-ps and OK1-ps-KA mutant expressed in Cos-1 cells. Histogram of the OK1-ps-KA mutant lane in the Phos-tag immunoblot and phosphorylation states (P0-P1) with band intensities (in%) are indicated. Please note dashed line indicates that the OK1-ps-KA mutant was run on the same gel but shown in higher exposure due to lower expression of the construct. (**d**) Modeling of the putative phosphorylation site within the obscurin inter-kinase region. The area included in the further analysis is highlighted by a dashed box, the phosphorylated serines identified by alignment with RyR2 are marked with asterisks (*). Residues contributing to the electropositive surface are shown in bold. Sequence predicted by C-Quark as helical is underlined and highlighted in the cartoon. Alignment with the homolog sequence in mouse obscurin-B and the resulting consensus sequence is shown below. Identical amino acids are highlighted in red. # = Asp or Glu. (**e**) Ab initio modeling of the putative autophosphorylation site within the inter-kinase region following human OK1. The peptide is shown modeled up to the first phosphorylated serine (Ser6789; shown as a sphere) using C-Quark. The surface of OK1 is colored in white. The OK1 homology model was structurally aligned to the following kinases and their inhibitory peptides and tails shown to illustrate possible binding locations of helical peptides on this family of kinases. Orange, inhibitory tail of twitchin kinase (pdb: 1kob [64]); light blue, inhibitory tail of calmodulin-dependent protein kinase (pdb: 1A06 [65]); lime green, inhibitory peptide of PKA1 (pdb: 1ATP [66]). (**f**) Electrostatic columbic surface of OK1 and its putative autophosphorylation substrate (rotated 180 degrees around its longitudinal axis compared to the peptide orientation shown in (**e**)) to show possible complementary interface. The surface has been colored by the charge of the residues it encloses; the surface is colored by a gradient of potential energy values ranging from −10 kcal/mol.e (red) to +10 kcal/mol.e (blue) (e is the charge of one electron).

**Figure 4 jcm-10-00984-f004:**
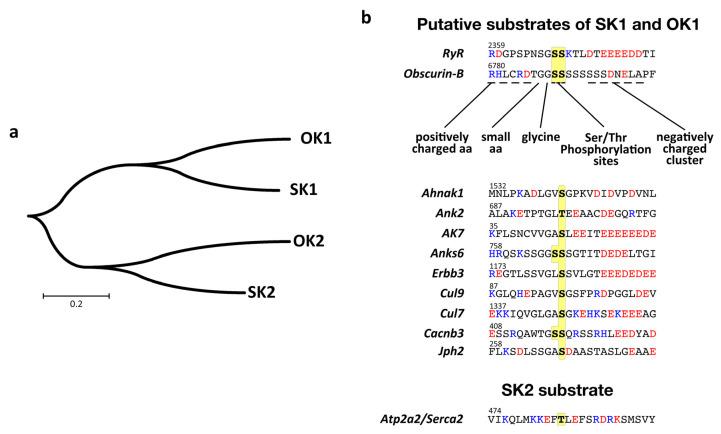
Putative substrate recognition and assignment for kinases 1 and 2 of SPEG and obscurin. (**a**) Molecular phylogenetic analysis of SPEG and obscurin kinases by the maximum-likelihood method indicates that the tandem kinase domains are more closely related between SPEG and obscurin than within each protein, suggesting that also phosphorylation site motifs should be similar between SK1 and OK1, as well as SK2 and OK2. (**b**) Analysis of phosphorylation site motifs of putative and known substrates for tandem SPEG and obscurin kinases. Only Serca2 has been in the literature unequivocally assigned as a substrate for SK2 [21]. Residues are colored according to charge (blue = positively charged; red = negatively charged; black = uncharged), with possible target serines/threonines highlighted in yellow.

**Table 1 jcm-10-00984-t001:** List of oligonucleotides used in this study.

Name	Sequence
mOK1.fwd	CCGCTCGAGCCACCATGGACAAGCTAGATGCCGAAAATCAAG
mOK1.rev	CGGGATCCGGTCTGGGGGACCCTGAAGCAGCTC
mOK1-ps.rev	CGGGATCCCGGGGCATGCTGGCCTCTGTCTCC
mOK1-ΔS7.rev	CGGGATCCCGGGCGCCACTGGCTTCCCCTCG
mSK1.fwd	CCGCTCGAGCCACCATGGAAGTCTCCTGCAAGGCGG
mSK1.rev	CGGGATCCGGGGGAGCCCGTAGCAGTTCC
mSK1-ps.rev	CGGGATCCGGGGTCCCCAGAGCCTCATCTTC
mOK1-KA.mut	GGAAACAAGATGTTCTGTGCCGCCGCCTTCATCCCCCTACGGAGTAAAAC

## Data Availability

The PDB-file of the homology model has been deposited to the Mendeley database and is accessible under the following doi:10.17632/pfmjz9s4pc.1.

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
