# Peer review of "Exploring Obscurin and SPEG Kinase Biology"

_jcm, 2021, doi:10.3390/jcm10050984_

Round 1

Reviewer 1 Report

The study by ? and colleagues focuses on the select group of proteins containing dual kinase activity, including obscurin and SPEG. Both proteins are abundantly expressed in striated muscles, where they play key roles in Ca regulation among other functions. Using biochemical evidence, molecular modeling and mutational analysis the authors provide novel evidence about the regulation of the more NH2-terminal kinase in obscurin via an autophosphosphorylation mechanism primarily involving the interkinase region without precluding the possibility of a Ca/CAM mediated regulation. Interestingly, such a mechanism (i.e. autophosphorylation) does not appear to regulate the activity of the NH2-terminal SPEG kinase. Moreover, using bioinformatics analysis a list of possible substrates for the NH2-terminal kinase in both proteins is suggested. Overall, the study is prudent and of interest to the field. Below please see specific comments to clarify and further improve the work included in the manuscript.

1. Given the results presented in Figure 1e and the presence of the P1 band, the conclusion of the authors that “the catalytic domain of OK1 is void of phosphorylation sites” (pg 7, lines 270-271) is questionable. Although the reviewer agrees that there are (auto)phosphorylation sites in the interkinase region of OK1, it is quite possible that there are (auto)phosphorylation sites within the kinase region, too. The authors should keep in mind that the OK1 and the OK1-DS7 proteins may not fold properly and that may affect their activity. The best control would be to mutagenize the series of Ser in the interkinase region in the OK1-ps construct and then perform phos-tag gel analysis in order to examine if there is (auto)phosphorylation within the kinase domain.

2. Fig. 2: “differentiated” C2C12 cells are still myoblasts and have not fully differentiated to myotubes containing striations; along these lines, the striations shown in the overlay in panel b cannot be seen in the relevant single-stain panel below. Thus, the conclusion that neither OK1 nor OK1-pS associate with sarcomeres is not supported. Moreover, do the authors suggest that phosphorylation of the interkinase region precludes OK1-pS from the nucleus in myotubes?

Author Response

Reviewer 1

The study focuses on the select group of proteins containing dual kinase activity, including obscurin and SPEG. Both proteins are abundantly expressed in striated muscles, where they play key roles in Ca regulation among other functions. Using biochemical evidence, molecular modeling and mutational analysis the authors provide novel evidence about the regulation of the more NH2-terminal kinase in obscurin via an autophosphosphorylation mechanism primarily involving the interkinase region without precluding the possibility of a Ca/CAM mediated regulation. Interestingly, such a mechanism (i.e. autophosphorylation) does not appear to regulate the activity of the NH2-terminal SPEG kinase. Moreover, using bioinformatics analysis a list of possible substrates for the NH2-terminal kinase in both proteins is suggested. Overall, the study is prudent and of interest to the field. Below please see specific comments to clarify and further improve the work included in the manuscript.

  1. Given the results presented in Figure 1e and the presence of the P1 band, the conclusion of the authors that “the catalytic domain of OK1 is void of phosphorylation sites” (pg 7, lines 270-271) is questionable. Although the reviewer agrees that there are (auto)phosphorylation sites in the interkinase region of OK1, it is quite possible that there are (auto)phosphorylation sites within the kinase region, too. The authors should keep in mind that the OK1 and the OK1-DS7 proteins may not fold properly and that may affect their activity. The best control would be to mutagenize the series of Ser in the interkinase region in the OK1-ps construct and then perform phos-tag gel analysis in order to examine if there is (auto)phosphorylation within the kinase domain.

We thank the reviewer for his/her comments. We agree that mutagenesis of the series of seven serines to alanine may clarify the autophosphorylation of the interkinase region C-terminal to OK1. We attempted this experiment, but were met with considerable technical problems due to the nature of the mutagenesis (a long stretch of 21 nucleotides encoding for the seven serine residues).

We concur with the reviewers comment that our truncation constructs may not fold properly, which in turn may affect their activity. We now address this concern in the discussion.

  1. Fig. 2: “differentiated” C2C12 cells are still myoblasts and have not fully differentiated to myotubes containing striations; along these lines, the striations shown in the overlay in panel b cannot be seen in the relevant single-stain panel below. Thus, the conclusion that neither OK1 nor OK1-pS associate with sarcomeres is not supported. Moreover, do the authors suggest that phosphorylation of the interkinase region precludes OK1-pS from the nucleus in myotubes?

We have now added the channel depicting endogenous obscurin with visible cross-striations as a single channel to the figure. The composite overlay in Figure 2b depicts a 4 channel image, HA in green (also shown as a separate channel in the bottom panels in a and b, F-actin in red, DAPI in blue and endogenous obscurin (using an antibody that recognizes obscurin isoforms A and B) in white. Sarcomeres that are visible using the obscurin antibody are not visible in the HA-channel that depicts the localization of the HA-OK1 and HA-OK1-ps constructs. We amended the figure legend and Figure 2 to highlight our findings and now include a panel that specifically depicts the sarcomeric localization of endogenous obscurin in a separate channel.

We hope that these changes help the reader to clarify this figure and figure legend.

Reviewer 2 Report

This paper investigates the functions of the kinase domains of Obscurin family proteins. It uses constructs of kinase1 domains and the interkinase region to determine if the kinase domain is a true kinase and whether autophosphorylation is a potential regulatory mechanism.   Extensive model studies enhance the significance of the findings.

Author Response

This paper investigates the functions of the kinase domains of Obscurin family proteins. It uses constructs of kinase1 domains and the interkinase region to determine if the kinase domain is a true kinase and whether autophosphorylation is a potential regulatory mechanism.   Extensive model studies enhance the significance of the findings.

We thank the reviewer for his/her evaluation.

Reviewer 3 Report

The paper by Fleming et al, entitled “Exploring Obscurin and SPEG kinase biology” aims to verify the presence of regulatory phosphorylation sites within the interkinase domain of SPEG and Obscurin-B . Through the phostag approach using the non-muscle COS-7 cells transfected with either SK1 or OK1 fragments, the authors found that only the Oscurin-B interkinase domain harbours at least three phosphorylation sites. They also present initial evidence of a potential autophosphorylation mechanism involving the catalytic subunit of OK1. Strengthened by  convincing bioinformatics analyses, this work provides new interesting insights into the understanding of the regulatory role of Obscurin in striated muscle tissue.

I only have some minor concerns:

1) All Obscurin and Speg fragments used in the experiments are generated from murine cDNAs. Alignments presented in figure 1a are from human database. To avoid to confound the reader it would be more appropriate to present the murine sequences and aa numbering integrated by the corresponding human aa numbering, and not vice versa, as presented.

2) In the phostag gel of figure 1c, the reported band corresponding to HA-SK1-ps (not phosphorylated) is detected with an apparent MW which does not alline to the SDS-page panel. The author should comment on that.

3) line 90-91. The authors state that “ loss of obscurin in mice in no apparent changes to cardiac morphology”. The indicated reference (31) is not appropriate as the data in that manuscript are reported as data not shown.

4) in the phostag gel of figure 3c the phosphorylated band of OK1-ps-KA lane is reported as P1 (one phosphorylated site), although it shares the same height as the P2 band of the OK1-ps lane. The authors should explain/correct.   

Author Response

The paper by Fleming et al, entitled “Exploring Obscurin and SPEG kinase biology” aims to verify the presence of regulatory phosphorylation sites within the interkinase domain of SPEG and Obscurin-B . Through the phostag approach using the non-muscle COS-7 cells transfected with either SK1 or OK1 fragments, the authors found that only the Oscurin-B interkinase domain harbours at least three phosphorylation sites. They also present initial evidence of a potential autophosphorylation mechanism involving the catalytic subunit of OK1. Strengthened by convincing bioinformatics analyses, this work provides new interesting insights into the understanding of the regulatory role of Obscurin in striated muscle tissue.

We thank the reviewer for his/her comments and suggestions.

I only have some minor concerns:

1) All Obscurin and Speg fragments used in the experiments are generated from murine cDNAs. Alignments presented in figure 1a are from human database. To avoid to confound the reader it would be more appropriate to present the murine sequences and aa numbering integrated by the corresponding human aa numbering, and not vice versa, as presented.

We thank the reviewer for his/her suggestion and altered the figure accordingly. We decided to keep the human sequence alignment in Figure 1a. This decision was done due to the fact that the manuscript reported the phosphorylation site in human RyR. However, we agree that it is confusing to first show the human sequence and then work with mouse constructs/proteins. Hence, another sequence alignment was added into Figure 1a that depicts the mouse versions of the proteins. We hope the alteration of the figure renders it more understandable.

2) In the phostag gel of figure 1c, the reported band corresponding to HA-SK1-ps (not phosphorylated) is detected with an apparent MW which does not alline to the SDS-page panel. The author should comment on that.

We thank the reviewer for the observation that the HA-SK1-ps bands in the phostag-gel and the standard SDS-page run slightly differently. This may have been caused by slightly different run-times of the standard and Phostag gels. However, it does not affect the conclusion of the experiment that HA-SK1 and HA-SK1-ps are not phosphorylated.

3) line 90-91. The authors state that “ loss of obscurin in mice in no apparent changes to cardiac morphology”. The indicated reference (31) is not appropriate as the data in that manuscript are reported as data not shown.

We agree that the manuscript refers to the data as ‘not shown’. We are currently working on a follow-up manuscript that includes a description of the cardiac phenotype of the obscurin knockout mice that corroborates this finding. Nevertheless, we changed the sentence to: “Surprisingly though, unlike SPEG knockouts loss of obscurin in mice is tolerable [31], suggesting that loss of function can be compensated during cardiac development.”

4) in the phostag gel of figure 3c the phosphorylated band of OK1-ps-KA lane is reported as P1 (one phosphorylated site), although it shares the same height as the P2 band of the OK1-ps lane. The authors should explain/correct.  

We agree with the reviewers comment on the migration of the phosphorylated band for OK1-ps-KA. However, we decided to keep the nomenclature as to denote the first identifiable phosphorylated band as P1, the second as P2 etc… While the OK1-ps-KA P1 band in Figure 3c is indeed at the approximate height of the P2 band in OK1-ps, we cannot exclude that the migration pattern may be influenced by other factors, such as position of the phosphorylation site within a protein or conformational changes (albeit being run in presence of SDS).

We thank the reviewer for his/her comments and hope that the changes render the manuscript acceptable in the eyes of this reviewer.

Reviewer 4 Report

The authors address an important problem of understanding domain functions of two similar giant proteins of the sarcomere, obscurin and SPEG. These proteins are known to be important to the proper assembly and organization of sarcomeres in both skeletal and cardiac muscle and likely play critical signaling functions. The domain complexity and large size of these proteins is a significant barrier to understanding function. The authors have chosen to study a very small region near the C-termini comprised of kinase domain 1 and a short region of the adjacent low sequence complexity inter-kinase domain linker in obscurin and SPEG.

  • Overall, while this study is reasonably well-documented and clearly written, I struggled to understand the objective of the work. This is evidenced by the non-descript title, and lack of clarity of Line 188 in the Introduction. Line 97 suggests that the paper will have to do with regulation of kinase domains through the inter-kinase regions and modulation of kinase activities but this information appears lacking in the paper. I strongly suggest editing the paper to focus the reader down on what the actual results are about (the inter-kinase regions function relative to the kinase function?). It might help to move some of the information in the Discussion to the Introduction and shorten some of the current sections of the Intro that are less related.
  • Using homology models and by knowing residues required for catalytic function, they show that both of the kinase domains in obscurin and in SPEG are probably capable of catalytic activity characteristic of serine kinases. However, as stated in the introduction, sequence identity alone suggests such a conclusion (lines 48-49) as well as prior experimental work (lines 54-55). Also their own experiments show that OK1 is an active kinase by comparing the native protein with a kinase dead mutation. I found this disconnect to be perplexing.
  • The most concrete conclusion of this paper is the fact that OK1 is a kinase and it can phosphorylate itself (although the exact residues were not determined). The modeling is a nice addition to show that the site is most likely an amphipathic helix similar to other known substrates and can fit in the expected pocket.
  • The obvious weakness of the paper is that while point 3) above is important, still nothing is known about auto-phosphorylation with regard to regulation of either the kinase itself or interactions with other substrates.
  • Within lines 416-429 in the Discussion the authors address the lack of any evidence for major phosphorylation sites within the catalytic domains of SK1 and OK1. The paragraph, which could be shortened, is accurate and honestly conveys the weaknesses of using only the COS-1 cells for the experiments. However, there is no explanation as to why they did not use C2C12 cells that they are already using for transfection experiments. Such data would be a significant addition to the paper.
  • The first paragraph in the Results section has nothing to do with the results. It should be deleted or integrated into the Introduction.
  • The analysis described starting at line 238 with regard to the alignments with a known RyR2 phosphorylation site is insightful and adds to the field. Start new paragraph on lines 245 and properly describe the experiments of COS cell transfection and expression. I suggest replacing the laborious description of the percentage band intensities by a small table that could be integrated into the figure or be a separate table that would also indicate which band is due to autophosphorylation by OK1.
  • Line 288 and paragraph describing the localization experiments. In my view, these data add nothing of significance to the paper for the obvious reasons that localization of small fragments of the proteins have no physiological meaning. If there was an assurance that the fragment localizing to the nuclei was phosphorylated, maybe. I am not convinced the nuclear localization is real, since a field of view is not shown. It is well known that staining protocols can disrupt nuclear membranes of cells.
  • Line 455, do the PKB studies suggest that your sites found are unlikely to be phosphorylated by PKB? There are very specific inhibitors of PKB that could potentially be useful in this context. There are other opportunities for pharmacological approaches in this study.

Minor points that must be addressed please.

The title should be revised to be more descriptive of the study.

It would be very helpful to refer to Figure 1 starting in the first paragraph of the Introduction and to make sure the figure includes annotations mentioned in the text when possible. For example, the PKB phosphorylation sites should be shown and constructs should be named as mouse in part b. Indicate the CaM binding site residue numbers for all proteins so they can be compared to the numbers corresponding to the constructs. I found it quite confusing that part a shows the human sequences but the constructs themselves are mouse, so I have little idea about actual sequence of your constructs. Line 264-266 is important but I have no way of knowing because of problem above. Paragraph starting at line 475 is hard to understand because of this deficiency. I do not have a good visual to help me understand where the autophos site is relative to the CaM site for example.

Please state that figures 1cde are images of western blots.

“Obscurin and SPEG kinases have been classified as myosin light chain kinases” line 45 is misleading suggesting that they phosphorylate the myosin light chain. Please restate.

Line 69, “The substantial amount of interaction partners…” should read number of interaction partners.

Line 95 “pathologically remodeling of the heart and arrhythmias” should be pathological

Line 119 “We identified that the inter-kinase region C-terminal to OK1” should be found not identified

Line 180, just want to make sure, this is Phos-tag small molecule not Phos-tag acrylamide? I think Phos-tag is correct?

Line 238 “of the two kinases within SPEG is responsible for this catalytic activity” should read two kinase domains

The statement on line 270 is inaccurate, you simply did not observe phosphorylation under these cell conditions that does not mean they do not contain phosphorylation sites.

Figure 2, perhaps I am dense, but I don’t understand colors of stains in Fig. 2b to see the white for obscurin. I am assuming you are using 4 colors total? Please clarify.

Line 381 see misspelling.

Line 384, why use the word degenerate?

Fig. 3e,f, since I missed the dashed box because it is so light the first time I looked (the “further analysis” reference was too obtuse for me), I kept wondering what the helix is that you are showing in e,f. Perhaps label positions of a few of the amino acids so we can compare it to part d.

Fig 3 F, “(f) Electrostatic columbic surface of OK1 and its putative autophosphorylation substrate (rotated 180 degrees around the longitudinal axis) to show possible complimentary interface. Rotated relative to what? Maybe outline the interfacial regions in both images? Or put arrows showing you have opened the sandwich?

Line 418, [58] suggest presence of at least one phosphorylation, insert “the”

Author Response

The authors address an important problem of understanding domain functions of two similar giant proteins of the sarcomere, obscurin and SPEG. These proteins are known to be important to the proper assembly and organization of sarcomeres in both skeletal and cardiac muscle and likely play critical signaling functions. The domain complexity and large size of these proteins is a significant barrier to understanding function. The authors have chosen to study a very small region near the C-termini comprised of kinase domain 1 and a short region of the adjacent low sequence complexity inter-kinase domain linker in obscurin and SPEG.

Overall, while this study is reasonably well-documented and clearly written, I struggled to understand the objective of the work. This is evidenced by the non-descript title, and lack of clarity of Line 188 in the Introduction. Line 97 suggests that the paper will have to do with regulation of kinase domains through the inter-kinase regions and modulation of kinase activities but this information appears lacking in the paper. I strongly suggest editing the paper to focus the reader down on what the actual results are about (the inter-kinase regions function relative to the kinase function?). It might help to move some of the information in the Discussion to the Introduction and shorten some of the current sections of the Intro that are less related.

We thank the reviewer for highlighting the lack of clarity in line 99 of the manuscript. We changed the sentence to “Few data are available that investigate the kinase domains in obscurin and SPEG, their biology, regulation and substrate specificity.”

We are not sure which statement the reviewer refers to when stating that line 188 lacks clarity. This line is located in the Materials and Methods section of the manuscript not the introduction.

Using homology models and by knowing residues required for catalytic function, they show that both of the kinase domains in obscurin and in SPEG are probably capable of catalytic activity characteristic of serine kinases. However, as stated in the introduction, sequence identity alone suggests such a conclusion (lines 48-49) as well as prior experimental work (lines 54-55). Also their own experiments show that OK1 is an active kinase by comparing the native protein with a kinase dead mutation. I found this disconnect to be perplexing.

We modified the sentence regarding the homology modelling at the end of the introduction to state: “Homology modelling substantiates that OK1 contains all key amino acids important for its catalytic activity in the right locations.”

We hope this resolves the disconnect between the perplexing statement and published data (lines 48-49; lines 54-55) and our results.

We also modified the sentence in the introduction that highlights the results from the homology modelling as follows: “Our homology modelling, mutational analysis and molecular docking demonstrate that kinase 1 in obscurin harbours all key amino acids important for its catalytic function, and that actions of this domain result in autophosphorylation of the protein.”

The most concrete conclusion of this paper is the fact that OK1 is a kinase and it can phosphorylate itself (although the exact residues were not determined). The modeling is a nice addition to show that the site is most likely an amphipathic helix similar to other known substrates and can fit in the expected pocket.

The obvious weakness of the paper is that while point 3) above is important, still nothing is known about auto-phosphorylation with regard to regulation of either the kinase itself or interactions with other substrates.

We thank the reviewer for pointing out that still nothing is “known about auto-phosphorylation with regard to regulation of either the kinase”. However, we believe that our studies will help inspire future experiments that investigate the regulation of kinase domain 1 in obscurin and SPEG by leading the way on a possible way to study the activity of these kinase domains.

Within lines 416-429 in the Discussion the authors address the lack of any evidence for major phosphorylation sites within the catalytic domains of SK1 and OK1. The paragraph, which could be shortened, is accurate and honestly conveys the weaknesses of using only the COS-1 cells for the experiments. However, there is no explanation as to why they did not use C2C12 cells that they are already using for transfection experiments. Such data would be a significant addition to the paper.

We agree with the reviewer that biochemical analysis of protein extracts from transfected C2C12 cells would be a significant addition to the manuscript. We attempted this experiment, however the transfection rate of C2C12 cells in our hands was too low (<1%) to allow for proper biochemical characterization using immunoblot analyses of standard and Phostag gels.

The first paragraph in the Results section has nothing to do with the results. It should be deleted or integrated into the Introduction.

We thank the reviewer for pointing this out and modified the results section accordingly.

The analysis described starting at line 238 with regard to the alignments with a known RyR2 phosphorylation site is insightful and adds to the field. Start new paragraph on lines 245 and properly describe the experiments of COS cell transfection and expression. I suggest replacing the laborious description of the percentage band intensities by a small table that could be integrated into the figure or be a separate table that would also indicate which band is due to autophosphorylation by OK1.

We agree with the reviewer and integrated most information on band intensities into the figures.

Line 288 and paragraph describing the localization experiments. In my view, these data add nothing of significance to the paper for the obvious reasons that localization of small fragments of the proteins have no physiological meaning. If there was an assurance that the fragment localizing to the nuclei was phosphorylated, maybe. I am not convinced the nuclear localization is real, since a field of view is not shown. It is well known that staining protocols can disrupt nuclear membranes of cells.

We appreciate the reviewers concerns regarding our staining protocol. Nevertheless, we would like to point out that undifferentiated C2C12 myoblasts transfected with the same construct did not show nuclear enrichment of HA-OK1, despite undergoing the same staining procedure. Hence, these cells may act as a control. We agree that small fragments may not necessarily reflect how full length proteins may act in the cellular context. However, the altered subcellular localization may be suggestive of changes occurring on the molecular level that are in our views worth pointing out.

Line 455, do the PKB studies suggest that your sites found are unlikely to be phosphorylated by PKB? There are very specific inhibitors of PKB that could potentially be useful in this context. There are other opportunities for pharmacological approaches in this study.

We thank the reviewer for this comment. The known PKB and CaMK sites in SPEG are C-terminal to the fragments we investigated. 

Minor points that must be addressed please.

The title should be revised to be more descriptive of the study.

We thank the reviewer for suggesting to change the title. We believe that our title, albeit being a little vague is reflecting the overall data presented in our manuscript. Specifically, data presented in Figures 1-3 investigate functions of kinase domains 1 in obscurin and SPEG, and highlights the autophosphorylation of the obscurin inter-kinase region, while Figure 4 also includes some data regarding kinase domain 2 in both proteins (the phylogenetic tree analysis and substrate described in the literature shown in alignment with known and putative substrates for kinase domain 1).

It would be very helpful to refer to Figure 1 starting in the first paragraph of the Introduction and to make sure the figure includes annotations mentioned in the text when possible. For example, the PKB phosphorylation sites should be shown and constructs should be named as mouse in part b.

We concur with the reviewers suggestion. We changed the figure to show identified phosphorylation sites in and highlight that the constructs shown in ‘panel b’ are of mouse origin by showing human aa-numbers in grey, as also pointed out in the modified figure legend.

Indicate the CaM binding site residue numbers for all proteins so they can be compared to the numbers corresponding to the constructs. I found it quite confusing that part a shows the human sequences but the constructs themselves are mouse, so I have little idea about actual sequence of your constructs.

We altered the figure to now depict both, the human and the mouse sequences. We decided to keep the human sequences, as the phosphorylation site in RyR2 was identified in humans, and there are some differences between the human and mouse sequences in SPEG and obscurin that are also highlighted in the Discussion section. However, we now specifically highlight also the CaM binding site in Figure 1a.

Line 264-266 is important but I have no way of knowing because of problem above. Paragraph starting at line 475 is hard to understand because of this deficiency. I do not have a good visual to help me understand where the autophos site is relative to the CaM site for example.

We agree with the reviewer and altered Figure 1 accordingly.

Please state that figures 1cde are images of western blots.

We thank the reviewer for his/her comment and altered the figure legends of Figure 1cde and Figure 3c to reflect that immunoblots are shown.

“Obscurin and SPEG kinases have been classified as myosin light chain kinases” line 45 is misleading suggesting that they phosphorylate the myosin light chain. Please restate.

We changed the sentence to “Obscurin and SPEG kinases have been classified to belong to the myosin light chain kinase family, while also showing considerable similarity to death-associated protein kinases (DAPKs; 58% similarity), titin kinase (54% similarity) as well as invertebrate twitchin (54% similarity) [7].”

Line 69, “The substantial amount of interaction partners…” should read number of interaction partners.

We changed the word according to the reviewers suggestion.

Line 95 “pathologically remodeling of the heart and arrhythmias” should be pathological

We thank the reviewer for pointing out this mistake and corrected it.

Line 119 “We identified that the inter-kinase region C-terminal to OK1” should be found not identified

We changed the text according to the reviewers suggestions.

Line 180, just want to make sure, this is Phos-tag small molecule not Phos-tag acrylamide? I think Phos-tag is correct?

It is Phostag-acrylamide. We apologize for the imprecise naming and amended the Materials and Methods section to reflect the correct substance.

Line 238 “of the two kinases within SPEG is responsible for this catalytic activity” should read two kinase domains

We changed the text according to the reviewers suggestions.

The statement on line 270 is inaccurate, you simply did not observe phosphorylation under these cell conditions that does not mean they do not contain phosphorylation sites.

We changed the statement as follows:

“In summary, the inter-kinase region following kinase domain 1 in obscurin contains several phosphorylation sites. Our data suggest also that the catalytic domains themselves (OK1 and SK1) are void of phosphorylation sites in the tested conditions.”

Figure 2, perhaps I am dense, but I don’t understand colors of stains in Fig. 2b to see the white for obscurin. I am assuming you are using 4 colors total? Please clarify.

We thank the reviewer for his/her comments. We amended the Figure and figure legend to clarify that ‘panel a’ is a 3 channel composite, while ‘panel b’ is a 4 layer composite image. In addition, we now also show the single channel magnified image for endogenous obscurin depicting cross-striations. We hope that the changes make it clearer to the reader what is shown.

Line 381 see misspelling.

We corrected the text according to the reviewers suggestions.

Line 384, why use the word degenerate?

We removed the word ‘degenerate’.

Fig. 3e,f, since I missed the dashed box because it is so light the first time I looked (the “further analysis” reference was too obtuse for me), I kept wondering what the helix is that you are showing in e,f. Perhaps label positions of a few of the amino acids so we can compare it to part d.

We thank the reviewer for this suggestion, which also led us to discover a mistake in the figure. We labeled the peptide and enhanced the color of dashed box shown in Figure 3e. We apologize for the error, and hope that the changes are acceptable.

Fig 3 F, “(f) Electrostatic columbic surface of OK1 and its putative autophosphorylation substrate (rotated 180 degrees around the longitudinal axis) to show possible complimentary interface. Rotated relative to what? Maybe outline the interfacial regions in both images? Or put arrows showing you have opened the sandwich?

We thank the reviewer for pointing out this lack in clarity. We amended the figure legend as follows: “(f) Electrostatic columbic surface of OK1 and its putative autophosphorylation substrate (rotated 180 degrees around its longitudinal axis compared to the peptide orientation shown in (e)) to show possible complimentary interface.”

Line 418, [58] suggest presence of at least one phosphorylation, insert “the”

We altered the text according to the reviewers suggestions.

We thank the reviewer for his/her thorough analysis of our manuscript, and appreciate the constructive comments and suggestions. We hope that the changes render the publication acceptable for publication.

Round 2

Reviewer 2 Report

This is a perfectly good manuscript describing the molecular biology of obscurin. It's actually quite novel and makes a significant contribution to a potentially important protein about which far too little is currently known. It hs been well revised.

The key defect is that it is totally unsuited to the Journal of Clinical Medicine since it contains no clinical and no medical content.  It seems to have been submitted to the wrong journal (IJMS would be very suitable) which is why I have declined to give a full report and recommended rejection by this Journal.

The conclusions of the study seem reasonable:  my major criticism is that the submitted manuscript does not fit the criteria for publication  in the Journal of Clinical Medicine.

Reviewer 4 Report

none